# Uncovering Functional Contributions of PMAT (*Slc29a4*) to Monoamine Clearance Using Pharmacobehavioral Tools

**DOI:** 10.3390/cells11121874

**Published:** 2022-06-09

**Authors:** Jasmin N. Beaver, Brady L. Weber, Matthew T. Ford, Anna E. Anello, Sarah K. Kassis, T. Lee Gilman

**Affiliations:** Department of Psychological Sciences & Brain Health Research Institute, Kent State University, Kent, OH 44242, USA; jbeave18@kent.edu (J.N.B.); bweber18@kent.edu (B.L.W.); mford27@kent.edu (M.T.F.); aanello@kent.edu (A.E.A.); skassis@kent.edu (S.K.K.)

**Keywords:** sex differences, psychostimulants, antidepressants, tail suspension test, locomotor activity, monoamine transporters, sensitization

## Abstract

Plasma membrane monoamine transporter (PMAT, Slc29a4) transports monoamine neurotransmitters, including dopamine and serotonin, faster than more studied monoamine transporters, e.g., dopamine transporter (DAT), or serotonin transporter (SERT), but with ~400–600-fold less affinity. A considerable challenge in understanding PMAT’s monoamine clearance contributions is that no current drugs selectively inhibit PMAT. To advance knowledge about PMAT’s monoamine uptake role, and to circumvent this present challenge, we investigated how drugs that selectively block DAT/SERT influence behavioral readouts in PMAT wildtype, heterozygote, and knockout mice of both sexes. Drugs typically used as antidepressants (escitalopram, bupropion) were administered acutely for readouts in tail suspension and locomotor tests. Drugs with psychostimulant properties (cocaine, D-amphetamine) were administered repeatedly to assess initial locomotor responses plus psychostimulant-induced locomotor sensitization. Though we hypothesized that PMAT-deficient mice would exhibit augmented responses to antidepressant and psychostimulant drugs due to constitutively attenuated monoamine uptake, we instead observed sex-selective responses to antidepressant drugs in opposing directions, and subtle sex-specific reductions in psychostimulant-induced locomotor sensitization. These results suggest that PMAT functions differently across sexes, and support hypotheses that PMAT’s monoamine clearance contribution emerges when frontline transporters (e.g., DAT, SERT) are absent, saturated, and/or blocked. Thus, known human polymorphisms that reduce PMAT function could be worth investigating as contributors to varied antidepressant and psychostimulant responses.

## 1. Introduction

Two broad classes of monoamine transporters regulate the amount and duration of extracellular monoaminergic signaling in the central nervous system (reviewed in [1,2,3,4]). These classes are known as uptake 1 (Na+ and Cl- dependent, high affinity, and low capacity), and uptake 2 (Na+ and Cl- independent, low affinity, and high capacity) [2,3,4,5,6,7,8,9]. Many psychoactive drugs, including those used to alleviate depression and anxiety symptoms (e.g., escitalopram, bupropion), as well as stimulants that are sometimes abused (e.g., cocaine, amphetamine), predominantly act upon uptake 1 transporters, including dopamine and serotonin transporters (DAT, *Slc6a3*; SERT, *Slc6a4*). Uptake 2 transporters, in contrast, lack selective inhibitors [3], making their functional contributions to monoaminergic signaling regulation challenging to study. Uptake 2 transporters are thought to function in a compensatory fashion when uptake 1 transporters are inhibited and/or overwhelmed with substrate, or in brain regions where uptake 1 transporters are minimally expressed [9]. Consequently, the contributions of uptake 2 transporters to monoaminergic signaling regulation are hypothesized to become prominent when uptake 1 transporters are inhibited by psychoactive drugs, such as escitalopram or cocaine. Thus, to circumvent the absence of selective uptake 2 inhibitors, we utilized mice with constitutive genetic reductions in the uptake 2 transporter plasma membrane monoamine transporter (PMAT; *Slc29a4*). In particular, we investigated how lifelong reductions in PMAT function influenced adult behavioral responses to psychoactive drugs that primarily inhibit uptake 1 transporters, particularly DAT and/or SERT. PMAT preferentially transports dopamine and serotonin over other monoamine neurotransmitters, albeit with approximately 400- and 600-fold lower affinity, respectively [10,11,12].

Little is known about how the functional loss of uptake 2 transporters influences responses to psychoactive drugs. The uptake 2 transporter organic cation transporter 3 (OCT3; *Slc22a3*) preferentially transports histamine, norepinephrine, and epinephrine over other monoamine neurotransmitters (e.g., dopamine, serotonin) [10]. A recent study focused on OCT3 provided *in vitro* and *in vivo* evidence suggesting that D-amphetamine mediates dopamine efflux via OCT3, though D-amphetamine is not a substrate for OCT3 [8]. Support for this conclusion is found in a recent report by Angenoorth and colleagues [13] indicating that *in vitro*, D-amphetamine inhibits PMAT with relatively low affinity (~72 μM), but does not inhibit OCT3. Investigations using conditioned place preference in mice suggest that both OCT3 and PMAT contribute to D-amphetamine-mediated reward, as indicated by time spent in previously D-amphetamine-paired chambers [14]. However, *in vivo* studies have thus far used only a single dose of D-amphetamine and have not explored any other psychoactive drugs (e.g., cocaine), nor non-stimulant drugs (e.g., escitalopram, bupropion) in mice with genetic reductions in an uptake 2 transporter. Moreover, though uptake 2 transporters as a whole are not well understood, PMAT remains particularly understudied among polyspecific cation transporters that transport monoamines within the brain [4].

To evaluate the behavioral effects of psychoactive drugs in PMAT-deficient mice, two different approaches were used. For non-stimulant psychoactive drugs (escitalopram and bupropion), mice were subjected to tail suspension and locomotor tests to assess how the drugs influenced antidepressant-predictive and locomotor behavior, respectively. For psychostimulants (cocaine and D-amphetamine), a psychostimulant-induced locomotor sensitization paradigm was used [15]. Overall, we hypothesized that mice with reduced (+/−) or ablated (−/−) PMAT function would exhibit augmented behavioral responses to psychoactive drugs, as compared to PMAT wildtype controls (+/+), given a diminished ability to compensate for pharmacologically impaired DAT/SERT function. Further, we anticipated sex-specific effects would be observed, with females exhibiting augmented sensitization to psychostimulants [16,17,18]. Sex-specific effects have been reported across PMAT genotypes as well [14,19,20], but given limited evidence in this realm, we did not have any *a priori* hypotheses regarding directionality of sex × genotype interactions for each drug.

## 2. Materials and Methods

### 2.1. Animals

Mice of both sexes were bred in-house and used for experiments at ≥90 days of age. Mice were the offspring of heterozygote × heterozygote (+/− × +/−) crosses, and were weaned at postnatal day 21, at which time ear punches were collected for mouse identification and genotyping. Mice constitutively deficient in PMAT were originally developed by the lab of Dr. Joanne Wang at the University of Washington [21]. A breeding colony was developed under a material transfer agreement (MTA) between Kent State University and the University of Washington. Breeding colonies originated from +/− PMAT-deficient mice on a C57BL/6J background shipped from the University of Texas Health Science Center at San Antonio (from the lab of Dr. Lynette C. Daws) to Kent State University, the latter with agreement from the University of Washington. Mice always had ad libitum access to LabDiet 5001 rodent laboratory chow (LabDiet, Brentwood, MO, USA) and water, and were housed in rooms maintained on a 12:12 light/dark cycle with lights on at 07:00. Mice were kept on 7090 Teklad Sani-chip bedding (Envigo, East Millstone, NJ, USA), and cages were changed weekly. Experiments were approved by the Institutional Animal Care and Use Committee at Kent State University and adhered to the National Research Council’s Guide for the Care and Use of Laboratory Animals, 8th Ed. [22].

### 2.2. Genotyping

Genomic DNA was extracted from ear punches using proteinase K (Roche, Basel, Switzerland), dissolved to 0.077% *w*/*v* on the day of extraction within a buffer of 100 mM Tris, 5 mM ethylenediaminetetraacetic acid (EDTA), 0.2% sodium dodecyl sulfate (SDS), and 200 mM NaCl, pH = 8.5 [20]. Genomic DNA (3.6 μL) was analyzed via PCR in 1X PCR buffer containing 1.74 mM MgCl2 and 34.7 μM dNTPs, with 0.20 μL of Platinum Taq (Invitrogen, Carlsbad, CA, USA) per 22 μL reaction. Amplification of the wildtype allele, between exons 3 and 4, and/or the knockout allele at the neomycin resistance cassette (Neo), was performed using 0.68 μL each of 10 μM primer stocks (Integrated DNA Technologies, Coralville, IA, USA) designed by Duan and Wang [21]: Exon 3 forward—5′ CGA CTA TCT TCA CCA CAA GTA CCC AG 3′; Exon 4 reverse—5′ GAG GCT CAT GTC AAA TAC GAT GGA G 3′; Neo F—5′ CTT GCT CCT GCC GAG AAA GTA TC 3′; and Neo R—5′ TCA GAA GAA CTC GTC AAG AAG GCG 3′. The following procedure was used for each PCR: 95 °C for 5 min; 34 cycles of 94 °C for 30 s, 59 °C for 30 s, and 72 °C for 90 s; 72 °C for 5 min; hold at 4 °C. To visualize PCR products, a 1% agarose gel electrophoresis was used. Gels were run at 150 V in 1X buffer of 40 mM Tris base, 0.114% acetic acid, and 1 mM EDTA, pH = 8.5, for 30 min. A 1 kb DNA ladder (Invitrogen) was used for DNA amplicon size reference, with the wildtype allele presenting at 847 bp, and the knockout allele at 447 bp [21]. Genotypes of experimental animals were re-verified post-mortem.

### 2.3. Drugs

All doses administered were calculated based on the salt form of each drug, except for escitalopram oxalate, which was calculated on the base form. Drugs were dissolved in sterile-filtered 0.9% NaCl (saline; vehicle). All injections were given intraperitoneal (ip.), at a volume of 10 mL/kg. Bupropion hydrochloride (PHR1730), escitalopram oxalate (E4786), cocaine hydrochloride (C5776), and D-amphetamine hemisulfate salt (A5880) were all purchased from Sigma Aldrich (St. Louis, MO, USA).

### 2.4. Behavior Tests

Females and males were always tested on different days. Mice were always moved in their home cages to the testing location at least 1 h prior to injection or test commencement, to permit acclimation. Testing always occurred during lights on. For every experimental paradigm, no more than one mouse per sex per genotype per drug treatment per litter was used, to minimize potential litter confounds.

#### 2.4.1. Tail Suspension Test (TST)

Mice were injected between 09:00 and 16:00, and tested 30 min after injection. Because no study to date has evaluated baseline behavior of PMAT-deficient mice in TST, we ran non-injected (i.e., naïve) mice alongside injected mice for TST experiments. Injected mice were treated with vehicle (saline), 1 or 2 mg/kg escitalopram, or 4 or 8 mg/kg bupropion [23,24]. The higher doses for each drug were selected because they were reported as being the lowest effective dose for each respective drug in TST, and the lower doses were half of these lowest effective doses, to test our hypothesis that mice with constitutive reductions in PMAT would exhibit greater behavioral responses to sub- or minimally effective doses of antidepressants in the TST. For testing, the tails of mice were gently secured to metal plates with 1” wide adhesive tape, and a loose cylindrical tube (41 mm L × 12 mm diameter) between the adhesive and the tail base, to prevent mice from climbing up their own tails and holding on to the plate during the test. Once tails were secured with tape to the metal plates, the metal plates were hung on hooks in the ceilings of separate, adjacent chambers for the 6 min test so that mice could not see each other during the test. Mice were oriented to be suspended so that their feet faced outwards to be visible to a video camera, which was used for recording behavior. After testing, mice were immediately released from the plates and tape and returned to their home cages. Offline scoring of TST behavior was performed by an observer blinded to mouse treatment and genotype. Time spent immobile and latency to first immobility for the 6 min test were scored using Solomon Coder (v. beta 19.08.02).

#### 2.4.2. Post-TST Locomotor Testing

Eight days after TST, mice that had already undergone TST were injected with a treatment different from that used for their TST, and tested for locomotor activity immediately after treatment injection for 1 h. This was done to minimize the number of mice used for experiments, in accord with the Three Rs for animal research [25]. We have previously reported on the lack of any locomotor activity differences between wildtype and PMAT-deficient mice in the absence of any injections [20], so we did not include a naïve condition for post-TST locomotor testing, again in accord with the Three Rs. Injections for locomotor testing were performed between 08:30–15:30. Arenas for locomotor testing were 45.7 H × 66.0 L × 38.1 cm W. Overhead cameras were used to record locomotor activity using ANY-Maze software (v. 7, Stoelting Co., Wood Dale, IL, USA). Distance traveled was quantified in 5 min bins, and the two bins concurrent with when the TST occurred post-injection (i.e., min 30–40) were analyzed to identify potential locomotor confounds when interpreting TST results. Locomotor activity for the entire 1 h duration of this test is presented in Appendix A, and accompanying statistics in Appendix A.

#### 2.4.3. Psychostimulant-Induced Locomotor Sensitization—Common Methods

Each injection day, mice were moved to the testing area at ~08:30 to acclimate to the environment and were weighed at that time. Experimental testing began at ~10:45 each injection day, starting with a 30 min habituation phase, followed immediately by a vehicle (saline) injection. Starting after the saline injection, testing occurred in 10 min bouts after each injection (1 saline injection + 4 drug injections; doses for individual drugs are detailed below). After the last test at the last dose for each injection day, mice were returned to their home cages and placed back in the colony until the next injection day. In addition to cumulative distance traveled under the influence of each psychostimulant, data were also graphed as percent change from same-sex and same-genotype cumulative distance traveled after drug given on Day 1, to more precisely assess each psychostimulant’s induction of locomotor sensitization across the subsequent four injection days.

##### 2.4.3.1. Cocaine-Induced Locomotor Sensitization

Following procedures optimized by Elliot [15], mice underwent a cocaine-induced locomotor sensitization paradigm every day for 5 consecutive days. Doses followed those of Elliot [15]: individual doses were 5, 5, 10, and 20 mg/kg, meaning cumulative doses of 5, 10, 20, and 40 mg/kg cocaine. Elliot demonstrated that this cumulative dosing paradigm produced a more robust locomotor sensitization to cocaine than single 40 mg/kg injections [15].

##### 2.4.3.2. D-Amphetamine Induced Locomotor Sensitization

Given that sensitization to amphetamine—unlike cocaine—is optimal in mice when there are drug-free days between injection days [26], mice were injected with amphetamine once every 3 days, for a total of 5 injection days. Individual doses were 0.1, 0.32, 1.0, and 3.2 mg/kg, meaning cumulative doses of 0.1, 0.42, 1.42, and 4.62 mg/kg [27,28].

### 2.5. Statistical Analyses

Figures were generated using GraphPad Prism 9.1.1 (GraphPad Software, San Diego, CA, USA), and statistical analyses were performed with GraphPad Prism and IBM SPSS Statistics 28.0.0.0 (IBM, Armonk, NY, USA). The significance threshold was set *a priori* at *p* < 0.05. During Day 1 of a cocaine experiment, after one female knockout was injected with her second 5 mg/kg cocaine dose, the camera for her arena failed, and thus data for this mouse was excluded due to an inability to calculate total distance traveled on Day 1, and percent change from Day 1. One male escitalopram 1 mg/kg knockout was more than six standard deviations outside the mean of his same-sex/genotype/treatment group for post-TST locomotor activity, and was thus excluded. Baseline conditions (naïve and saline-treated mice in TST, saline-treated mice in post-TST locomotor, and Day 1 cumulative psychostimulant-induced locomotor responses were analyzed with a 2-way ANOVA (genotype × sex) and Holm–Šídák post hoc testing to identify directional effects of PMAT deficiency. All other TST, post-TST locomotor, and psychostimulant-induced locomotor sensitization data were analyzed using a 3-way ANOVA (treatment × genotype × sex, or day × genotype × sex) and pairwise comparisons with Bonferroni correction. For within-subjects analyses, Greenhouse–Geisser corrections were utilized. Data were graphed as the mean ± the standard error of the mean (SEM), or as violin plots showing individual data points plus medians and quartiles.

## 3. Results

### 3.1. TST Behavior

Mice constitutively deficient in PMAT have not previously been tested in TST, so we evaluated behavioral responses to TST in the absence of any injections (i.e., naïve). In these mice, there was no interaction between genotype × sex in time spent immobile during TST (F (2,42) = 1.42, *p* = 0.252, partial η^2^ = 0.064) (Figure 1A). There was also no effect of sex (F (1,42) = 1.99, *p* = 0.166, partial η^2^ = 0.045), although a non-significant trend for genotype was noted (F (2,42) = 2.45, *p* = 0.099, partial η^2^ = 0.104). Holm-Šídák’s post hoc testing revealed that, relative to male wildtype mice, male heterozygotes (*p* = 0.0414) and knockouts (*p* = 0.0414) exhibited significantly increased time immobile in TST, whereas no differences relative to same-sex wildtypes were observed in females (*p* > 0.8). No significant genotype × sex interaction in latency to first immobility bout in TST was detected (F (2,42) = 0.802, *p* = 0.455, partial η^2^ = 0.037), nor were main effects of sex (F (1,42) = 1.88, *p* = 0.177, partial η^2^ = 0.043) or genotype (F (2,42) = 0.564, *p* = 0.573, partial η^2^ = 0.026) (Figure 1B). In contrast to naïve mice, when evaluating time immobile during TST in saline-injected mice, no significant genotype × sex interaction was observed (F (2,43) = 0.507, *p* = 0.606, partial η^2^ = 0.023) (Figure 1C). Likewise, main effects of sex (F (1,43) = 0.089, *p* = 0.767, partial η^2^ = 0.002) and genotype (F (2,43) = 0.737, *p* = 0.485, partial η^2^ = 0.033) were not observed. However, while no significant genotype × sex interaction (F (2,43) = 0.700, *p* = 0.502, partial η^2^ = 0.032) nor main effect of sex (F (1,43) = 0.796, *p* = 0.377, partial η^2^ = 0.018) were observed in saline-injected mice for latency to first immobility bout, a significant genotype effect was found (F (2,43) = 3.548, *p* = 0.037, partial η^2^ = 0.142) (Figure 1D). Holm-Šídák’s post hoc testing indicated that female saline-injected knockout mice exhibited significantly shorter latencies to first immobility bout relative to female saline-injected wildtype mice (*p* = 0.0291).

To facilitate interpretation of TST behavioral changes in response to drug injections, and to control for the behavioral effects of injection stress, data were graphed and analyzed as percent change from same-sex and same-genotype saline-treated mice, as done previously [24]. Graphs are separated by sex for clarity, but analyses were performed across sexes. Starting with percent change in TST immobility time relative to same-sex and same-genotype mice injected with saline, there was no three-way interaction of treatment × genotype × sex (F (8,212) = 0.925, *p* = 0.496, partial η^2^ = 0.034), nor was there a treatment × genotype interaction (F (8,212) = 0.676, *p* = 0.713, partial η^2^ = 0.025). There were, however, significant interactions of genotype × sex (F (2,212) = 3.574, *p* = 0.030, partial η^2^ = 0.033) and treatment × sex (F (4,212) = 2.605, *p* = 0.037, partial η^2^ = 0.047) (Figure 2A). Pairwise comparisons indicated that in male wildtypes (*p* < 0.001) and heterozygotes (*p* = 0.028), mice treated with 2 mg/kg escitalopram exhibited significantly less immobility relative to same-sex and same-genotype saline-injected controls (Figure 2C). Moreover, the response of male wildtypes to 2 mg/kg escitalopram was significantly different from the response of female wildtypes to the same drug and dose (*p* < 0.001; Figure 2C). For percent change in latency to the first immobility bout during TST relative to same-sex and -genotype mice injected with saline, once again, no three-way interaction of treatment × genotype × sex (F (8,212) = 1.412, *p* = 0.193, partial η^2^ = 0.051) was observed. Unlike with percent change in TST immobility, no interaction of treatment × sex occurred (F (4,212) = 1.673, *p* = 0.157, partial η^2^ = 0.031). A non-significant trend for a treatment × genotype interaction was noted (F (8,212) = 1.866, *p* = 0.067, partial η^2^ = 0.066). As with percent change in TST immobility, a genotype × sex interaction was significant (F (2,212) = 8.720, *p* < 0.001, partial η^2^ = 0.076). There was also a significant main effect of treatment (F (4,212) = 11.157, *p* < 0.001, partial η^2^ = 0.174). Pairwise comparisons revealed that, relative to saline-injected female knockouts, female knockouts injected with either dose of escitalopram or the higher 8 mg/kg dose of bupropion exhibited significantly greater latencies to the first immobility bout (all *p* < 0.001; Figure 2B). Further, female knockouts injected with 1 mg/kg (*p* < 0.001) or 2 mg/kg (*p* = 0.025) escitalopram, or 8 mg/kg bupropion (*p* = 0.009) displayed enhanced percent changes in latencies to the first immobility bout relative to male knockouts injected with the same drug and dose (Figure 2B).

### 3.2. Post-TST Locomotor Behavior

To evaluate the possibility that interpretations of TST results might be confounded by locomotor-induced changes, mice that underwent TST testing were subsequently tested 8 days later for locomotor behavior following a different injection treatment than what the mouse received for TST. We have previously reported that there are no overall locomotor changes from PMAT deficiency alone [20], so unlike the protocol used for TST behavior, we did not include a naïve group here. We specifically analyzed locomotor behavior occurring between 30–40 min post-injection, corresponding to the same time frame as when TST testing occurred (Figure 3). Locomotor data for the entire duration of the 1 h test are presented in the Appendix A. Saline-injected mice similarly did not exhibit any significant interaction between genotype × sex (F (2,47) = 0.977, *p* = 0.384, partial η^2^ = 0.040), nor were main effects of genotype (F (2,47) = 0.028, *p* = 0.972, partial η^2^ = 0.001) or sex (F (1,47) = 1.608, *p* = 0.211, partial η^2^ = 0.033) observed (Figure 3A). Holm-Šídák’s post hoc testing likewise did not indicate any differences (all *p* > 0.58). Subsequently, as for TST data, we graphed and analyzed post-TST locomotor behavior as a percent change from saline-injected mice of the same sex and genotype, and for clarity, the sexes are graphed separately. A non-significant trend was noted for treatment × genotype × sex (F (8,233) = 1.728, *p* = 0.093, partial η^2^ = 0.056). While there was not a significant treatment × genotype interaction (F (8,233) = 1.163, *p* = 0.322, partial η^2^ = 0.038), there were significant interactions between treatment × sex (F (4,233) = 3.079, *p* = 0.017, partial η^2^ = 0.050) and genotype × sex (F (2,233) = 22.366, *p* < 0.001, partial η^2^ = 0.161). Pairwise comparisons indicated that in female wildtypes, mice treated with either dose of escitalopram or the higher 8 mg/kg bupropion dose exhibited significantly greater locomotor activity relative to saline-injected female wildtypes (all *p* = 0.002) (Figure 3B). In female knockouts, only treatment with 2 mg/kg escitalopram significantly increased locomotor activity relative to saline-treated controls (*p* = 0.048). Male heterozygotes only exhibited elevated locomotor activity in response to either dose of bupropion (both *p* < 0.001) relative to saline-injected controls (Figure 3C). Across sexes for each genotype, there were several sex differences in response to drug treatment. Female wildtypes exhibited significantly elevated locomotor activity in response to 1 mg/kg (*p* = 0.002) or 2 mg/kg (*p* < 0.001) escitalopram relative to male wildtypes. In contrast, male heterozygotes displayed greater locomotor activity in response to 4 mg/kg (*p* = 0.031) or 8 mg/kg (*p* < 0.001) bupropion in comparison to female heterozygotes. Locomotor responses to 1 mg/kg (*p* = 0.027) or 2 mg/kg (*p* = 0.003) escitalopram were higher in female knockouts compared to male knockouts (Figure 3B,C).

In addition to behavioral responses to the non-stimulant drugs escitalopram and bupropion, we also investigated how constitutive PMAT deficiency affected psychostimulant-induced locomotor sensitization using a cumulative dosing paradigm. First, we explored locomotor sensitization to cocaine, which inhibits uptake by DAT, SERT, and the norepinephrine transporter (NET) [29]; then we investigated locomotor sensitization to D-amphetamine, a DAT substrate that induces the DAT- and OCT-mediated efflux of dopamine [8].

### 3.3. Cocaine-Induced Locomotor Sensitization

The total distance traveled under the influence of cocaine on Day 1 did not result in a genotype × sex interaction (F (2,44) = 0.506, *p* = 0.607, partial η^2^ = 0.022) (Figure 4A). No main effect of genotype was observed (F (2,44) = 0.357, *p* = 0.702, partial η^2^ = 0.016), but there was a main effect of sex (F (1,44) = 5.91, *p* = 0.019, partial η^2^ = 0.118).

When evaluating total distance travelled across the 5 consecutive days of injections, there was no three-way interaction of day × genotype × sex (F (5.28,116.0) = 0.175, *p* = 0.975, partial η^2^ = 0.008) (Figure 4B,C), nor two-way interactions of genotype × sex (F (2,44) = 1.089, *p* = 0.346, partial η^2^ = 0.047) or day × genotype (F (5.28,116.0) = 0.411, *p* = 0.850, partial η^2^ = 0.018). While there was a significant two-way interaction of day × sex (F (2.64,116.0) = 2.973, *p* = 0.041, partial η^2^ = 0.063), there was no main effect of genotype (F (2,44) = 0.039, *p* = 0.962, partial η^2^ = 0.002).

When cocaine data were normalized to a percentage of Day 1 for each sex-genotype combination, analyses revealed no three-way interaction of day × genotype × sex (F (4.73,104.0) = 0.546, *p* = 0.731, partial η^2^ = 0.024) (Figure 4D,E). As expected based on previous studies [16,17,18], there was a significant interaction between day × sex (F (2.36,104.0) = 10.502, *p* < 0.001, partial η^2^ = 0.193). No interactions of day × genotype (F (4.73,104.0) = 1.48, *p* = 0.207, partial η^2^ = 0.063) or genotype × sex (F (2,44) = 0.234, *p* = 0.792, partial η^2^ = 0.011) were detected. Likewise, there was no main effect of genotype (F (2,44) = 2.27, *p* = 0.115, partial η^2^ = 0.094). Pairwise comparisons indicated that on Day 5, female heterozygotes exhibited significantly less (*p* = 0.039) locomotor sensitization as compared to same-sex wildtypes (Figure 4D).

### 3.4. Amphetamine-Induced Locomotor Sensitization

On Day 1 of amphetamine experiments, there was no genotype × sex interaction (F (2,41) = 2.074, *p* = 0.139, partial η^2^ = 0.092), nor were there main effects of genotype (F (2,41) = 0.207, *p* = 0.814, partial η^2^ = 0.010) or sex (F (1,41) = 0.410, *p* = 0.526, partial η^2^ = 0.010) (Figure 5A). When analyzing amphetamine-induced locomotor activity across days, there was not a significant three-way interaction of day × genotype × sex (F (5.68,116.5) = 0.941, *p* = 0.465, partial η^2^ = 0.044) (Figure 5B,C). We did observe an expected day × sex interaction (F (2.84,116.5) = 4.031, *p* = 0.010, partial η^2^ = 0.090), but there was not a day × genotype interaction (F (5.68,116.5) = 0.628, *p* = 0.699, partial η^2^ = 0.030). A trend towards significance was noted for genotype × sex (F (2,41) = 2.875, *p* = 0.068; partial η^2^ = 0.123); for clarity, sexes are graphed separately (Figure 5B,C). There was no main effect of genotype (F (2,41) = 0.257, *p* = 0.775, partial η^2^ = 0.012). Pairwise comparisons found that female heterozygotes had significantly less amphetamine-induced locomotor activity on the third day of injections (Day 7) as compared to female wildtypes (*p* = 0.029) (Figure 5B).

As with cocaine, amphetamine data were normalized to Day 1 distance traveled for the same sex and genotype, to better evaluate drug sensitization over time. When evaluating the data in this manner, a non-significant trend was observed for day × genotype × sex (F (5.34,109.5) = 2.041, *p* = 0.074, partial η^2^ = 0.091) (Figure 5D,E). Given that this analysis normalizes to Day 1 for the same sex and genotype, it is not surprising that there was no significant day × sex interaction (F (2.67,109.5) = 1.516, *p* = 0.218, partial η^2^ = 0.036). There was also no significant day × genotype interaction (F (5.34,109.5) = 0.915, *p* = 0.479, partial η^2^ = 0.043), but a trend towards a genotype × sex interaction was noted (F (2,41) = 2.495, *p* = 0.095, partial η^2^ = 0.108). An expected main effect of day was detected (F (2.67,109.5) = 186.4, *p* < 0.001, partial η^2^ = 0.82), but main effects of sex (F (1,41) = 0.912, *p* = 0.345, partial η^2^ = 0.022) and genotype (F (2,41) = 0.610, *p* = 0.548, partial η^2^ = 0.029) were not significant.

## 4. Discussion

Across psychoactive compounds, our results indicate that PMAT function is sexually dimorphic, a revelation that required perturbations in monoaminergic signaling either via pharmacological mechanisms or by an acute stressor. These findings agree with previous reports that behavioral and physiological consequences of PMAT deficiency emerge in a sex-specific manner following homeostatic perturbations [19,20]. Moreover, the outcomes observed align with current thinking that PMAT is engaged in a compensatory manner, recruited when uptake 1 transporters are saturated and/or incapacitated but otherwise remaining relatively quiescent, and/or exists as a substitute monoamine uptake mechanism in brain regions where uptake 1 transporter expression is scant (e.g., cerebellum, frontal cortex) [3,9]. Still, our results did not align with our hypotheses about PMAT-deficient mice in many instances, both with our anticipation of enhanced behavioral responses to the non-stimulant compounds escitalopram and bupropion, as well as with our hypothesis that locomotor sensitization to cocaine and D-amphetamine would be augmented. Though portions of each of these expected outcomes were supported by some data to varying extents, nuances of sex, genotype, drug, dose, and day all contributed to create a much more complex story than PMAT ‘merely’ serving as a catch-all for the castoffs of uptake 1 transporters.

Constitutive deficiency in PMAT sex-selectively affected TST immobility behavior in naïve males, with both heterozygous and knockout males displaying increased immobility, while no genotype effect was observed across genotypes in naïve females. In contrast, and highlighting the importance of evaluating behavior in non-injected animals, TST immobility was unaffected across sex and genotype in saline-injected mice. When assessing latency to the first immobility bout in TST, no genotype nor sex effects were observed in naïve mice, but saline-injected female knockouts exhibited a reduced latency relative to saline-injected female wildtypes, whereas no differences were observed across genotypes in latency to first immobility in saline-injected males. Thus, the experience of a saline injection stress was sufficient to affect behavioral responses to a different brief stressor—that of the TST—and to both obscure (male PMAT-deficient) and elicit (female PMAT knockout) sex- and genotype-specific responses.

Administration of non-stimulant drugs escitalopram and bupropion further emphasized sex- and genotype-specific effects, though not exactly in the manner we hypothesized. Administration of the higher dose of escitalopram attenuated TST immobility in male wildtype mice, similar to previous work [23,24]. However, this effect was ablated in PMAT knockout male mice, in direct contrast to our hypothesis. Moreover these drugs, at both their lower and higher doses, elicited no changes in TST immobility in female wildtype mice, highlighting how doses optimized for the male sex [23] do not always translate to the female sex. Unlike in males, where no changes in latency to the first immobility bout occurred after drug administration, female PMAT knockouts exhibited significantly increased latencies in response to both doses of escitalopram, as well as to the higher dose of bupropion. Unlike what we observed with immobility times, these data align with our hypothesis, but in a sex-selective manner. Collectively, our findings indicate that intact PMAT function could sex-selectively counteract specific behavioral changes elicited by escitalopram and bupropion in females, but facilitate other behavioral changes evoked by these drugs in males.

When assessing an activity-related measure such as immobility, considerations of potential confounds like broader effects on locomotor behavior are important. In agreement with our previous work showing that PMAT deficiency does not impact locomotor activity in non-injected mice [20], we observed that saline injections did not significantly affect locomotor behavior across genotypes in both sexes. An unexpected observation was that all but the lower dose of bupropion enhanced locomotor activity in female wildtypes, yet these mice exhibited no significant shifts in TST immobility. Female knockouts displayed increased locomotor activity only in response to the higher dose of escitalopram, suggesting the heightened latency to first immobility in this specific sex, genotype, and treatment group might be confounded by enhanced locomotion. In males, the locomotor-enhancing effects of both bupropion doses were specific to heterozygotes, but as with female wildtypes, there curiously was no significant change in TST measures in bupropion-treated male heterozygotes. Combined, these data can be interpreted to suggest that—with the exception of female knockouts given 2 mg/kg escitalopram—either there are likely not locomotor confounds in TST behavior, or alternatively, that the locomotor effects of these drug-dose-genotype-sex combinations potentially obscured reductions in TST immobility (or augmentations in latency).

Regardless, the TST and locomotor data together provide strong evidence for PMAT deficiency eliciting sex-specific behavioral responses to psychoactive drugs that inhibit SERT (escitalopram) or DAT and NET (bupropion). Researchers in several labs [3,13] have reported that bupropion has weak inhibitory action (~100 μM) at PMAT *in vitro*, and that (es)citalopram does not act at PMAT at all. In our study, bupropion’s effects were significant in female knockouts (TST latency), female wildtypes (locomotor activity), and male heterozygotes (locomotor activity). Given the relatively low (4 and 8 mg/kg) doses of bupropion we employed here, these effects are most likely unmasking the contribution of PMAT uptake under conditions when DAT/NET function is impaired, rather than any (lack of) action at PMAT (see also [30]). Likewise, the behavioral influences of escitalopram in male wildtypes (TST immobility), female wildtypes (locomotor activity), male heterozygotes (TST immobility), and female knockouts (TST latency and locomotor activity) illustrate how PMAT likely facilitates the uptake of monoamines when SERT function is blocked. For example, our TST data suggest that intact PMAT function facilitates the lowered immobility induced by escitalopram in males, whereas in females PMAT probably compensates for impaired SERT function due to escitalopram blockade by keeping extracellular serotonin levels relatively low. Certainly, neurochemical investigations using microdialysis or voltammetric techniques would be necessary to investigate these possibilities.

Given the relatively modest effects of non-stimulant psychoactive drugs in PMAT-deficient mice, we next pursued a more heavy-handed pharmacological approach to investigate how cumulative dosing of psychostimulants that act at DAT/SERT/NET (cocaine) or primarily DAT and NET (D-amphetamine) affected locomotor sensitization. We anticipated that repeated dosing with these more behaviorally activating drugs would elicit clearer sex- and PMAT genotype-specific behavioral responses, but once again, our findings did not support this hypothesis. Initial (Day 1) locomotor responses to either cocaine or D-amphetamine revealed no influence of PMAT deficiency, nor any suggestion of sex as a moderator. Moreover, although we observed the anticipated augmented sensitization to both psychostimulants in females compared to males, PMAT genotype did not reliably impact locomotor sensitization to either drug. Instead, we observed quite modest attenuations in cocaine- and D-amphetamine-induced locomotor sensitization only on a single day for each drug (day 5 and day 7, respectively), and only in female heterozygotes. There was also a non-significant trend (*p* < 0.10) for male knockouts over the final three days of D-amphetamine-induced locomotor sensitization. Once again, these investigations illustrate how there appears to be a relationship between sex and PMAT deficiency that only emerges after stress or uptake 1 inhibition, but further studies are necessary to determine the mechanisms responsible for this relationship. Multiple reports agree that cocaine does not act at PMAT, but evidence is more conflicted regarding amphetamine, with the Sitte lab providing evidence that amphetamine has some action at hPMAT at concentrations of ~72 uM *in vitro* [3,4,13]. Our cocaine findings align with these reports, and the eventual attenuation of cocaine-induced locomotor sensitization in female PMAT heterozygotes could indicate that PMAT’s role in psychostimulant-induced sensitization is secondary and may be obscured in female knockouts due to constitutive compensatory changes. Indeed, the potential minor involvement of PMAT in sensitization processes to uptake 1-acting psychostimulants, at least in females, is suggested by our data. An alternative interpretation, and one not mutually exclusive with the preceding statement, is that the doses of these psychostimulants did not generate extracellular monoamine levels sufficient enough to robustly reveal the entire contribution of PMAT to monoamine uptake, despite these doses being literature-based [15,27,28]. One other report has observed a modest relationship between PMAT deficiency and amphetamine-mediated locomotor sensitization [14]; though this investigation used a single 1 mg/kg dose administered over four injection days to either wildtypes or knockouts, in contrast to the cumulative dosing paradigm over five injection days given to all three genotypes here.

Unlike the Day 1 psychostimulant locomotor responses, our TST and locomotor results following escitalopram and bupropion administration suggest PMAT deficiency might sex-selectively influence antidepressant drug responses. The effectiveness of any given antidepressant treatment in humans is notoriously unpredictable [31,32,33]. This unpredictability has been attributed, among other things, to genetic polymorphisms in uptake 1 transporters [34,35,36]. Functional polymorphisms in the human PMAT gene (*SLC29A4*) can affect treatment responses to metformin, a drug for type II diabetes management [37,38,39]. However, studies have yet to evaluate how *SLC29A4* polymorphisms might be associated with antidepressant treatment response in humans, or indeed any measurements of overall mood or other mental states. This is likely a consequence, at least in part, of candidate gene investigations falling out of favor in the literature despite lingering evidence that some polymorphism findings are replicable [40,41,42]. Polymorphisms in *SLC29A4*, such as rs3889348 [37,39], substantially reduce PMAT function, meaning heterozygous PMAT mice could serve as a model for attenuated PMAT function in humans.

Indeed, much remains unexplored about PMAT function across species. The absence of a selective pharmacological inhibitor of PMAT is a prominent roadblock to such studies, reflecting the necessity for the indirect approach here with uptake 1 inhibitors in combination with constitutive genetic deficiency of PMAT. Compensatory development-specific and/or lifelong upregulation of similar transporters is certainly an inherent concern when making conclusions with such rodent models. The Wang lab, which developed these PMAT-deficient mice, reported that mRNA for SERT, DAT, NET, and OCT3 were unaffected in the adult brains of these mice, allaying such concerns [21]. Still, the compensatory protein expression of these transporters remains a potential limitation, as does the possibility that PMAT could influence brain development, given that PMAT mRNA has been detected throughout mouse embryo brains [9]. The sex-specific effects observed here and previously [20], for instance, might instead be explained by compensatory upregulation of OCT3, which unlike PMAT, can be inhibited by sex hormones such as progesterone and estradiol (see review in [3]). Alternatively, and not mutually exclusive of the preceding possibility, is that stress hormone release in response to injections inhibits OCT3—which could be compensatorily upregulated in PMAT-deficient mice—thereby inducing the blockade of OCT3 and muddying interpretations of the contributions of PMAT alone (see review by [4]).

Our current understanding of the function of uptake 2 transporters is that they predominantly serve as a backup/substitute system for uptake 1 transporters [9]. Consequently, compensatory lifelong upregulation of other transporters may be unlikely. Indeed, it is the very nature of these uptake 2 transporters as backups that appears to make their constitutive absence challenging to detect until monoaminergic systems are sufficiently stimulated, such as in adult acute stress situations like TST. The brief stress of TST unmasked the behavioral consequence of reduced PMAT function specifically in naïve males, as revealed by their increased immobility behavior, suggesting intact PMAT function in males might facilitate active stress coping behaviors [43,44,45]. Additional studies are necessary to identify how PMAT function contributes to heterotypic stressor responsivity and to determine if the sexually dimorphic effects of PMAT function are organizational or activational. Inclusion of heterozygotes in future investigations will be crucial, given their biological mirroring of recognized functional human PMAT polymorphisms. This is particularly relevant considering that the present findings support an underappreciated role for PMAT function in sex-specific responses to drugs used in humans to influence mood, cognition, anxiety, attention, and other mental states.

## Figures and Tables

**Figure 1 cells-11-01874-f001:**
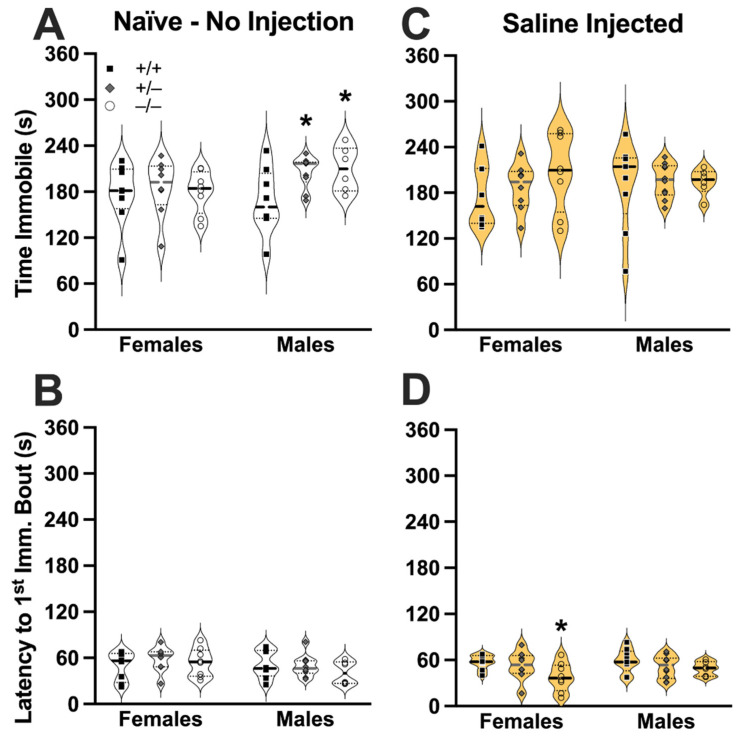
**Naïve and saline-injected mouse behavior in tail suspension test.** Naïve (i.e., non-injected; clear) female, and male PMAT wildtype (+/+, black squares), PMAT heterozygote (+/−, grey diamonds), and PMAT knockout (−/−, open circles) mice underwent the tail suspension test (TST), and measures of time spent immobile (**A**) and latency to the first bout of immobility (**B**) were determined offline by an observer blinded to treatment and genotype. Likewise, saline-treated (yellow, 10 mL/kg) PMAT mice underwent TST 30 min after injection, and time spent immobile (**C**) and latency to the first immobility bout (**D**) were quantified in the same manner as naïve mice. Data are shown as individual points in violin plots, with horizontal lines indicating median and quartiles. * *p* < 0.05 vs. same-sex, same treatment PMAT wildtype mice.

**Figure 2 cells-11-01874-f002:**
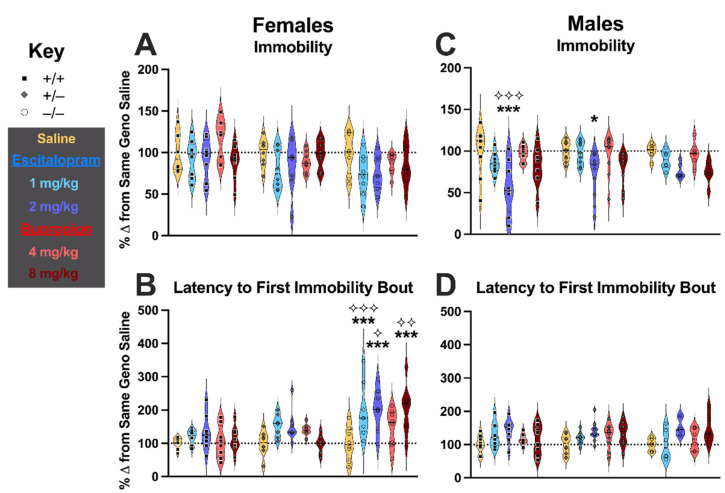
**Escitalopram- and bupropion-injected mouse behavior in tail suspension test.** Behavior in the tail suspension test (TST) was normalized to same-sex and same-genotype saline-injected mice (yellow, 10 mL/kg) to best evaluate how injections of escitalopram (light blue, 1 mg/kg; dark blue, 2 mg/kg) or bupropion (light red, 4 mg/kg; dark red, 8 mg/kg) affected TST behavior in female (**A**,**B**) and male (**C**,**D**) PMAT wildtype (+/+, black squares), PMAT heterozygote (+/−, grey diamonds), and PMAT knockout (−/−, open circles) mice. Data were graphed as percent change from the same-sex and same-genotype saline-injected mice for time spent immobile (**A**,**C**) and latency to the first bout of immobility (**B**,**D**), based on scoring offline by an observer blinded to treatment and genotype. Data are shown as individual points in violin plots, with horizontal lines indicating median and quartiles. The dashed line across all graphs indicates the mean of 100% for saline-injected controls. * *p* < 0.05, *** *p* < 0.001 vs. same-sex, same-genotype saline-injected mice. ⟡ *p* < 0.05, ⟡⟡ *p* < 0.01, ⟡⟡⟡ *p* < 0.001 vs. opposite-sex, same-genotype, same treatment mice.

**Figure 3 cells-11-01874-f003:**
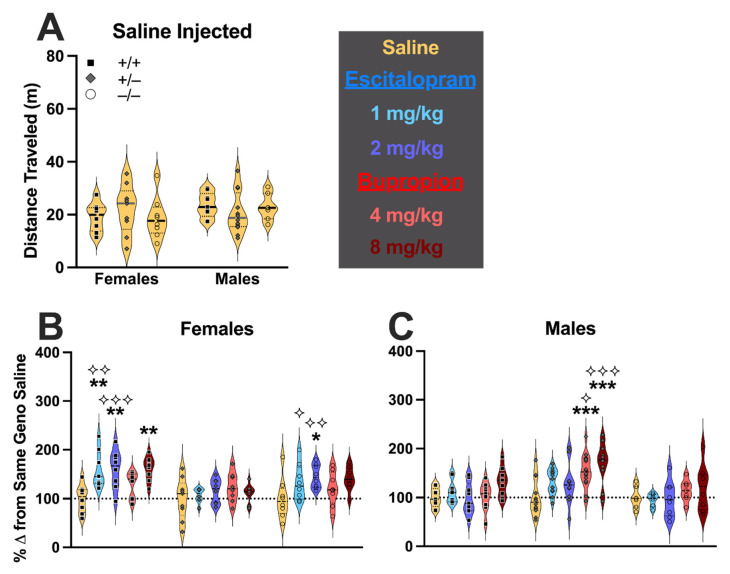
**Escitalopram- and bupropion-injected mouse locomotor behavior.** Locomotor behavior in the open field was not different across sex nor genotype in saline-injected (yellow, 10 mL/kg) PMAT wildtype (+/+, black squares), PMAT heterozygote (+/−, grey diamonds), and PMAT knockout (−/−, open circles) mice (**A**). Locomotor data were normalized to same-sex and same-genotype saline-injected mice to best evaluate how injections of escitalopram (light blue, 1 mg/kg; dark blue, 2 mg/kg) or bupropion (light red, 4 mg/kg; dark red, 8 mg/kg) affected locomotor behavior in female (**B**) and male (**C**) PMAT mice. Data were graphed as percent change from the same-sex and same-genotype saline-injected mice for distance traveled, as quantified by ANY-maze software. Data are shown as individual points in violin plots, with horizontal lines indicating median and quartiles. The dashed line across (**B**,**C**) indicates the mean of 100% for saline-injected controls. * *p* < 0.05, ** *p* < 0.01, *** *p* < 0.001 vs. same-sex, same-genotype saline-injected mice. ⟡ *p* < 0.05, ⟡⟡ *p* < 0.01, ⟡⟡⟡ *p* < 0.001 vs. opposite-sex, same-genotype, same treatment mice.

**Figure 4 cells-11-01874-f004:**
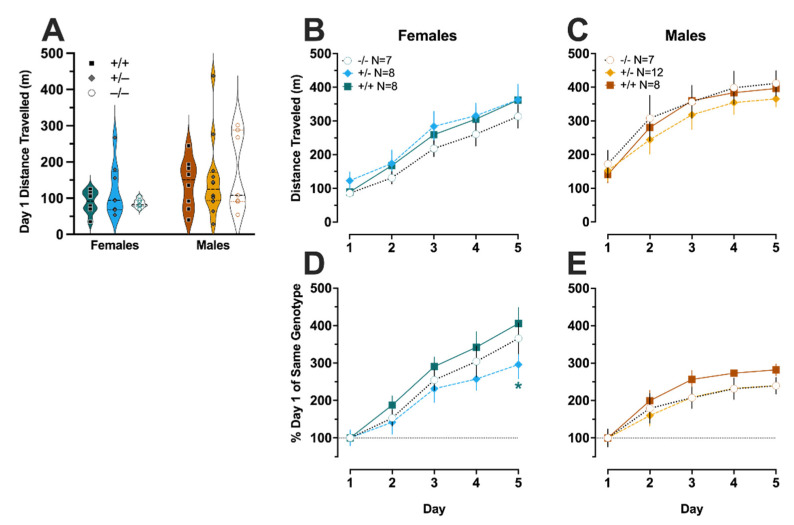
**Cocaine-induced locomotor behavior and locomotor sensitization.** Locomotor behavior in the open field was not different across sex nor genotype in PMAT wildtype (+/+, black squares), PMAT heterozygote (+/−, grey diamonds), and PMAT knockout (−/−, open circles) mice (**A**) in response to a cumulative dose of 40 mg/kg (individual injections of 5, 5, 10, and 20 mg/kg cocaine, each 10 min apart) on Day 1 of 5 total injection days. Over 5 consecutive injection days, locomotor data in response to a cumulative dose of 40 mg/kg cocaine each day were graphed across days for females (**B**) wildtypes, green squares; heterozygotes, blue diamonds; knockouts, white circles) and males (**C**) wildtypes, orange squares; heterozygotes, yellow diamonds; knockouts, white circles). These same locomotor data in response to cocaine were also normalized to Day 1 data for same-sex and same-genotype mice, to best evaluate how repeated cocaine exposure over 5 consecutive days elicited cocaine-induced locomotor sensitization in female (**D**) and male (**E**) PMAT mice. Data were graphed as percent change from Day 1 cocaine-induced locomotion for same-sex and same-genotype mice, as quantified by ANY-maze software. Data in (**A**) are shown as individual points in violin plots, with horizontal lines indicating median and quartiles. Data in (**B**–**E**) are graphed as means ± SEM. The dashed line across (**D**,**E**) indicates the mean of 100% for Day 1 cocaine-induced locomotion for the same-sex and same-genotype. * *p* < 0.05 vs. same-sex wildtype mice on same day.

**Figure 5 cells-11-01874-f005:**
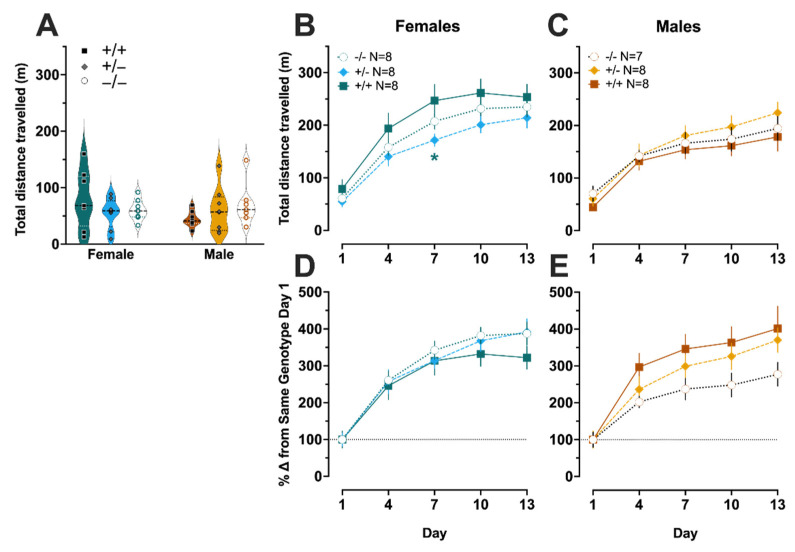
**D-amphetamine-induced locomotor behavior and locomotor sensitization.** Locomotor behavior in the open field was not different across sex nor genotype in PMAT wildtype (+/+, black squares), PMAT heterozygote (+/−, grey diamonds), and PMAT knockout (−/−, open circles) mice (**A**) in response to a cumulative dose of 4.62 mg/kg (individual injections of 0.1, 0.32, 1.0, and 3.2 mg/kg D-amphetamine, each 10 min apart) on Day 1 of 5 total injection days. Over 5 injection days, each separated by 3 days, locomotor data in response to a cumulative dose of 4.62 mg/kg D-amphetamine each day were graphed across days for females (**B**) wildtypes, green squares; heterozygotes, blue diamonds; knockouts, white circles) and males (**C**) wildtypes, orange squares; heterozygotes, yellow diamonds; knockouts, white circles). These same locomotor data in response to D-amphetamine were also normalized to Day 1 data for same-sex and same-genotype mice, to best evaluate how repeated D-amphetamine exposure over 5 injection days elicited D-amphetamine-induced locomotor sensitization in female (**D**) and male (**E**) PMAT mice. Data were graphed as percent change from Day 1 D-amphetamine-induced locomotion for same-sex and same-genotype mice, as quantified by ANY-maze software. Data in (**A**) are shown as individual points in violin plots, with horizontal lines indicating median and quartiles. Data in (**B**–**E**) are graphed as means ± SEM. The dashed line across (**D**,**E**) indicates the mean of 100% for Day 1 D-amphetamine-induced locomotion for the same-sex and same-genotype. * *p* < 0.05 vs. same-sex wildtype mice on same day.

## Data Availability

The data presented in this study are available in the Appendix A.

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
