# Peer review of "Uncovering Functional Contributions of PMAT (Slc29a4) to Monoamine Clearance Using Pharmacobehavioral Tools"

_cells, 2022, doi:10.3390/cells11121874_

Round 1
Reviewer 1 Report
The manuscript of Beaver et al investigates the role of PMAT in monoamine clearance, in vivo. The authors use PMAT knock-out mice and basic behavioral assays to understand if, upon pharmacological manipulation of SERT and DAT, PMAT can contribute to the effects elicited by antidepressants and psychostimulants. The study also considers and founds sex differences in these effects which increases the relevance of the study
The manuscript reads well and provides in vivo evidence of the role of PMAT in monoamine clearance when DAT and SERT are saturated or inhibited, which is of interest to the transporter community. However, to strengthen the authors’ conclusions additional work is required:
1. The behavioral tests are all conducted 30 min after i.p. injection. This ignores the possibility that PMAT could participate in the pharmacokinetics of the drugs studied. Preliminary information regarding this aspect could be derived by re-analyzing the open field experiments using 5/10 minutes bins and showing the kinetics of the behavioral experiments over 60 min.
2. The study seems underpowered. Was a power analysis performed to determine the animal group? this can also be conducted post-hoc to confirm if the actual number of mice used is sufficient to consider statistical significance relevant. Despite not ideal, if the group size will result too small the number of mice should be increased accordingly to avoid the misinterpretation of the data.
3. Figure 3: panels are too small – consider changing the figure layout to 2x2 to increase the size of the figure
4. Figure 4: why cumulative inj have been conducted instead of a single injection? An explanation should be provided. Moreover, cocaine has quite a fast pharmacokinetics: after the 4th inj (20mg/Kg) the first dose (5mg/Kg) might be already metabolized therefore not reaching the desired final dose (40mg/Kg).
1. Important references should be also included and discussed
B Bönisch H. Substrates and Inhibitors of Organic Cation Transporters (OCTs) and Plasma Membrane Monoamine Transporter (PMAT) and Therapeutic Implications. Handb Exp Pharmacol. 2021;266:119-167. doi: 10.1007/164_2021_516. PMID: 34495395.
Maier J et al. The Interaction of Organic Cation Transporters 1-3 and PMAT with Psychoactive Substances. Handb Exp Pharmacol. 2021;266:199-214. doi: 10.1007/164_2021_469. PMID: 33993413.
Discussion should then be revised according to the new experiments requested.
Reviewer 2 Report
Manuscript "Uncovering functional contributions of PMAT (Slc29a4) to 2 monoamine clearance using pharmacobehavioral tools" by J. N. Beaver et al.
In this manuscript, the authors were interested in understanding the contribution of the plasma membrane monoamine transporter (PMAT, Slc29a4) to drugs that selectively block DAT/SERT and their behavioral consequences. For this purpose, they tested Slc29a4+/+, Slc29a4+/- and Slc29a-/- mice of both sexes in response to acute antidepressants (escitalopram, bupropion) administration in tail suspension test and locomotion or to psychostimulant (cocaine, D-amphetamine) in repeated administration to assess locomotor sensitization.
The authors report that male Slc29a4+/- and Slc29a-/- mice exhibited significantly increased immobility time in TST, whereas no differences relative to same-sex wildtypes were observed in females. Saline-injected female Slc29a4-/- mice exhibited significantly shorter latencies to first immobility bout relative to saline-injected female Slc29a4+/+ mice. In response to antidepressant, male Slc29a4+/+ and Slc29a4+/- mice treated with 2 mg/kg escitalopram exhibited significantly less immobility relative to same-sex and -genotype saline-injected controls and male Slc29a4+/+ mice response was significantly different from the response of female Slc29a4+/+ mice. Relative to saline-injected, female Slc29a4-/- mice injected with either dose of escitalopram or 8 mg/kg dose of bupropion exhibited significantly greater latencies to first immobility bout, and enhanced changes in latencies to first immobility bout relative to male Slc29a4-/- mice injected with the same drug and dose. Female Slc29a4+/+ mice treated with either dose of escitalopram or the higher 8 mg/kg bupropion dose exhibited significantly greater locomotor activity relative to saline-injected female Slc29a4+/+ mice. In female Slc29a4-/- mice, only treatment with 2 mg/kg escitalopram significantly increased locomotor activity relative to saline-treated controls. Male Slc29a4+/- mice exhibited elevated locomotor activity in response to either dose of bupropion relative to saline-injected controls, and displayed greater locomotor activity in response to 4 mg/kg or 8 mg/kg bupropion in comparison to femaleSlc29a4+/- mice. Finally, in response to repeated injection of cocaine, female Slc29a4+/- mice exhibited, only on day 5, significantly less locomotor sensitization as compared to female Slc29a4+/+ mice. Furthermore, female Slc29a4+/- mice had significantly less amphetamine-induced locomotor activity on day 7 as compared to female Slc29a4+/+ mice.
The authors conclude that PMAT acts differently across sexes, and that likely PMAT’s monoamine clearance contribution emerges when high affinity transporters (e.g., DAT, SERT) are saturated and/or blocked.
The contribution of PMAT to antidepressant and psychostimulant action is indeed of interest. These manuscript experiments are sound but generated little positive results. This manuscript could gain in general interest if completed by complementary experiments.
In particular, the authors are correct in saying that "the experience of a saline injection stress was sufficient to affect behavioral responses to a different brief stressor – that of the TST – and both obscure (male PMAT-deficient) and elicit (female MAT knockout) sex- and genotype-specific responses" or "The brief stress of TST unmasked the behavioral consequence of reduced PMAT function specifically in naïve males, as revealed by their increased immobility behavior, suggesting intact PMAT function in males might facilitate active stress coping behaviors". These observations should be completed by experimentally assessing the stress response of these mice by performing a real stress experiment and evaluating stress response, including corticosterone levels.
Evaluation of steroid hormones in these mice would also be interesting for interpreting the sex differences.
One missing information is the localization of PMAT in the mouse brains, which has been described (Dahlin et al., Neuroscience 2007 vol. 146 (3) pp. 1193-211) and shows the expression of PMAT in regions where DAT or SERT are poorly expressed, which makes rather difficult the interpretation that PMAT is "recruited when uptake 1 transporters are saturated and/or incapacitated". In addition, this paper shows quite strong PMAT expression during embryonic development, which could have developmental consequences in constitutive knockout mice and affect the interpretation of the results. This has at least to be discussed.
The authors are correct in saying that "neurochemical investigations using microdialysis or voltammetric techniques would be necessary to investigate these possibilities". Evaluating the monoamine levels in response to acute antidepressant or repeated psychostimulant would bring a lot to interpret the results.
It is incomplete to say in the abstract and introduction that PMAT "transports monoamine neurotransmitters, including dopamine and serotonin, faster than more studied monoamine transporters, e.g., dopamine transporter (DAT), serotonin transporter (SERT)". This statement should be completed by saying that the affinity of PMAT for serotonin or dopamine are about 1000-fold lower than for these other transporters.
Round 2
Reviewer 1 Report
The Authors have clearly and sufficiently answered my concerns. I recommend the revised manuscript for publication.